# TARS 🪙: MINMAX TOKEN-ADAPTIVE PREFERENCE STRATEGY FOR MLLM HALLUCINATION REDUCTION

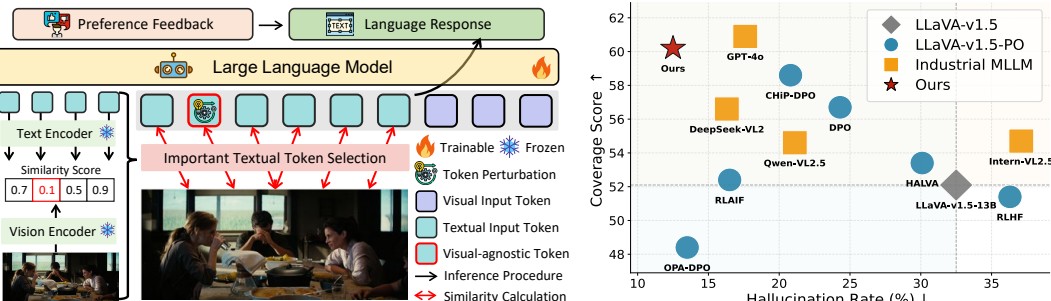

Figure 1: **Left:** We present *TARS*, a token-adaptive preference strategy for mitigating hallucinations in MLLMs. TARS reformulates direct preference optimization (DPO) as a min-max objective that (1) minimizes behavioral misalignment via preference feedback and (2) maximizes adaptability through perturbations of visual-agnostic tokens. **Right:** Evaluation on LLaVA-v1.5-13B with preference optimization (PO) (Liu et al., 2023b) and various MLLMs under AMBER benchmark (Wang et al., 2023) shows that TARS surpasses PO baselines and matches GPT-4o (Hurst et al., 2024).

## ABSTRACT

Multimodal large language models (MLLMs) enable vision-language reasoning, yet often generate plausible outputs that are factually incorrect or visually ungrounded, thereby compromising their reliability. Direct preference optimization (DPO) is a common strategy for correcting hallucinations by aligning model outputs with human preferences. However, existing DPO strategies typically treat hallucination-related preferences as fixed targets, relying on static and potentially biased supervision signals during training. This approach tends to overfit to superficial linguistic cues in preference data, leading to distributional rigidity and spurious correlations that impair grounding in causally relevant visual information. To overcome this limitation, we propose TARS, a token-adaptive preference strategy that reformulates DPO as a min–max optimization problem. TARS maximizes token-level distributional shifts under explicit semantic constraints to simulate alignment uncertainty, and simultaneously minimizes the expected preference loss under these controlled perturbations. This joint objective effectively preserves causal grounding while mitigating overfitting to preference patterns, thereby reducing hallucinations in multimodal reasoning. We evaluate TARS on multiple hallucination benchmarks and find consistently strong and robust performance. Using only 4.8k preference samples and no expert feedback, TARS reduces hallucination rates from 26.4% to 13.2% and decreases cognition value from 2.5 to 0.4, outperforming standard DPO and matching GPT-4o on several key metrics.

## 1 INTRODUCTION

Large language models (LLMs) demonstrate strong reasoning capabilities across a broad range of language tasks (Gandhi et al., 2023; Chen et al., 2023; Wang et al., 2025). Building on this foundation, multimodal large language models (MLLMs) integrate visual inputs to enable grounded understanding and vision-language reasoning (Tong et al., 2024; Huang et al., 2023; Driess et al., 2025). Although this integration broadens their applicability in various tasks (Jiang et al., 2024b; Shao et al., 2025), it also introduces key challenges, among which hallucinations are particularly

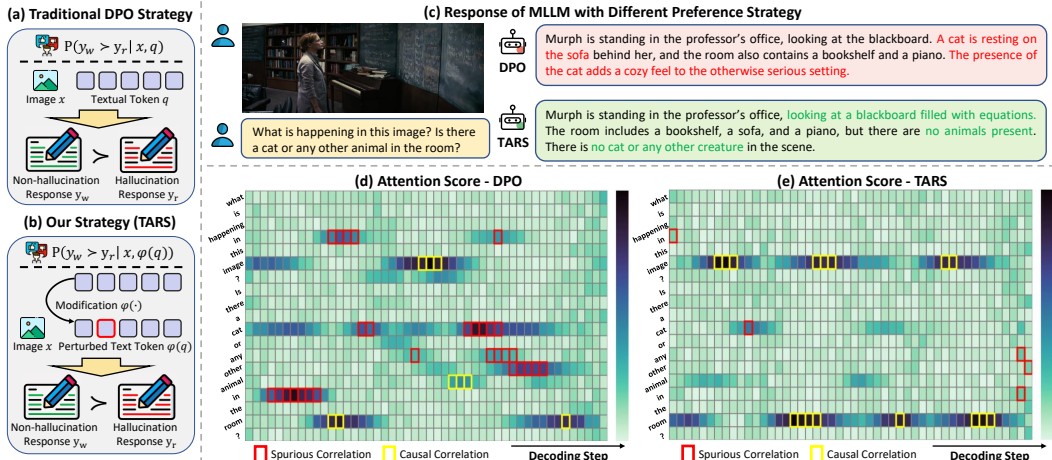

Figure 2: Motivation illustration for TARS. (a) and (b) illustrate standard DPO and our token-adaptive strategy. (c) shows a VQA example where DPO hallucinates, while TARS avoids ungrounded output. (d) and (e) visualize token-to-query attention maps during decoding. DPO over-attends to spurious tokens, while TARS attends to causally grounded visual-semantic cues.

prominent (Kim et al., 2024; Jiang et al., 2024a; Huang et al., 2024). Hallucinations in MLLMs refer to outputs that may appear plausible yet are factually incorrect or lack grounding in the visual context (Sarkar et al., 2025b; Gunjal et al., 2024). Addressing these failures is essential for improving the reliability, safety, and practical applicability of MLLMs in real-world applications.

Modern MLLMs are typically developed through a two-stage training pipeline that includes a knowledge-intensive pretraining phase (Dai et al., 2023; Bao et al., 2022; Zhang et al., 2024), followed by instruction tuning (Liu et al., 2024a; Wang et al., 2024b; Liu et al., 2023a). These stages endow the model with broad world knowledge and the ability to follow instructions in natural language (Bao et al., 2022; Liu et al., 2023b). Despite these capabilities, hallucinations often stem not from knowledge deficits, but from behavioral biases acquired during training that lead the model to generate plausible yet ungrounded outputs (Oh et al., 2024; Chen et al., 2025). To address these failures, preference optimization (PO) has become a prominent strategy for reducing hallucinations by aligning model outputs with human expectations (Schulman et al., 2017; Achiam et al., 2017). PO fine-tunes models using ranked response pairs derived from either human feedback (Sun et al., 2024; Ouyang et al., 2022) or AI-generated preferences (Yu et al., 2025; Sharma et al., 2024), providing direct supervision to reinforce grounded and faithful responses. Such methods have proven effective in mitigating hallucinations across diverse tasks.

Direct preference optimization (DPO) (Rafailov et al., 2023) has become a widely adopted method for hallucination reduction (Fu et al., 2025; Yang et al., 2025). Current DPO methods rely heavily on preference data, which can cause models to overfit to shallow textual cues, such as high-frequency phrases or repetitive patterns in the training set (Huang et al., 2024; Liu et al., 2024b). Prior studies have observed that this overfitting leads MLLMs to generate responses that appear plausible but lack visual grounding, as illustrated in Figure 2(c). In our analysis (Figure 2(d)), we further find that DPO-trained models often assign high preference to outputs containing spurious correlation tokens, including prepositions or frequently mentioned objects, even when these elements are not visually grounded (Xie et al., 2024; Wang et al., 2024a). These observations reveal a **core limitation** of DPO: its reliance on static preference signals hinders generalization under shifting visual–textual contexts, leading to brittle cross-modal alignment and increased vulnerability to hallucination (Setlur et al., 2024; Fu et al., 2025). This rigidity also prevents the model from adapting to local semantic discrepancies, weakening its ability to ground responses in causally relevant visual information.

We formulate the challenge of distributional rigidity in preference optimization as a **min-max token-adaptive alignment problem**: maximizing distributional variation under semantic constraints, followed by minimizing the expected preference loss under these controlled perturbations. Specifically, we introduce perturbations to visual-agnostic tokens, textual elements with minimal cross-modal grounding, to simulate contextual variation and shift the input distribution without altering the semantic content. This setup enables the model to rely on causally grounded visual signals rather

than superficial textual correlations, thereby mitigating hallucinations arising from overfitting to preference data (see Figure 2(b)). We refer to this approach as **TARS** (token-adaptive preference strategy), a lightweight and generalizable approach that enhances preference learning by introducing distribution-aware variability during fine-tuning.

To validate the effectiveness of our method, we evaluate TARS on LLaVA-v1.5 (Liu et al., 2023b) at 7B and 13B scales, comparing it against leading preference optimization approaches. Across a comprehensive suite of hallucination benchmarks spanning generative and discriminative tasks, TARS-enhanced LLaVA-v1.5 achieves consistently strong performance and matches GPT-4o (Hurst et al., 2024) in several settings. These results underscore the effectiveness of token-adaptive preference optimization in reducing hallucinations and advancing the trustworthiness of MLLMs.

Our contributions are as follows:

- We reformulate preference learning as a min-max optimization objective that encourages token-level distributional shifts within semantic boundaries, while minimizing preference misalignment, thereby mitigating overfitting to rigid or spurious supervision signals.
- We introduce **TARS**, a lightweight strategy that perturbs visual-agnostic tokens to simulate semantic variation, enhancing visual grounding and reducing hallucinations.
- TARS achieves SOTA hallucination reduction using only 4.8k preference samples and no expert feedback, matching GPT-4o on several key metrics.

## 2 PRELIMINARIES

**Multimodal Large Language Models.** MLLMs extend LLMs by incorporating visual inputs alongside textual prompts (Zhang et al., 2024). Formally, given an image $x$ and a prompt $q$, the model generates a textual response $y = (y_1, \ldots, y_l)$ in an autoregressive manner (Liu et al., 2023b):

$$y_t \sim \pi_\theta(y_t \mid y_{<t}, x, q), \tag{1}$$

where $\pi_\theta$ denotes the conditional generation policy parameterized by $\theta$. Given a textual input $q$ and a visual input $x$, the model tokenizes them into discrete sequences: textual tokens $q = \{q_1, \ldots, q_m\}$ and visual tokens $x = \{x_1, \ldots, x_n\}$. These tokens are mapped to embeddings and fused via cross-attention to integrate semantic signals from both modalities. The resulting context is then used by the decoder to autoregressively generate the output sequence (Dou et al., 2022; Yang et al., 2021).

**Direct Preference Optimization.** Direct preference optimization (DPO) (Rafailov et al., 2023) is an effective approach for aligning model behavior with human preferences. It bypasses explicit reward models by directly optimizing preferences from pairwise comparisons.

Traditional methods such as reinforcement learning with human feedback (RLHF) (Ouyang et al., 2022) and AI feedback (RLAIF) (Yu et al., 2025) rely on training a scalar reward model $r_\psi(x, q, y)$ from preference pairs. This reward model is typically trained using the Bradley-Terry formulation (Bradley & Terry, 1952):

$$\begin{aligned} P\left(y_w \succ y_r \mid x, q\right) &= \frac{\exp(r_\psi(x, q, y_w))}{\exp(r_\psi(x, q, y_w)) + \exp(r_\psi(x, q, y_r))} \\ &= \sigma\left(r_\psi(x, q, y_w) - r_\psi(x, q, y_r)\right), \end{aligned} \tag{2}$$

where $(x, q, y_w, y_r)$ is sampled from the preference data distribution $\mathcal{D}$, $\sigma(z) = \frac{1}{1+\exp(-z)}$ denotes the sigmoid function. $y_w$ and $y_r$ denote the preferred and dispreferred responses, respectively. $r_\psi(x, q, y)$ is trained to maximize the log-likelihood of correctly ranking the preferred response:

$$\min_{r_\psi} \mathbb{E}_{(x,q,y_w,y_r)\sim\mathcal{D}} \left[-\log \sigma\left(r_\psi(x, q, y_w) - r_\psi(x, q, y_r)\right)\right], \tag{3}$$

After training, the learned reward model $r_\psi(x, q, y)$ is used to guide the fine-tuning of the policy $\pi_\theta$. Specifically, the policy is optimized to generate high-reward responses while minimizing divergence from a fixed reference policy $\pi_{\text{ref}}$, typically using KL-regularized objectives:

$$\min_{\pi_\theta} \mathbb{E}_{(x,q)\sim\mathcal{D},\, y^*\sim\pi_\theta(y^*|x,q)} \left[-\left(r_\psi(x, q, y^*) - \alpha \cdot \mathbb{D}_{\text{KL}}\left(\pi_\theta(y^* \mid x, q) \,\|\, \pi_{\text{ref}}(y^* \mid x, q)\right)\right)\right], \tag{4}$$

Figure 3: Overview of **TARS**. TARS reformulates preference optimization as a Min–Max problem: (1) The maximization branch perturbs visual-agnostic tokens to simulate semantically shifted contexts (red dashed box); (2) The minimization branch fine-tunes the model to align with human preferences via the DPO objective (purple dashed box). TARS encourages the model to attend to causally grounded visual signals rather than spurious correlations, thereby reducing hallucinations.

where $\alpha$ controls the strength of regularization, which ensures alignment with the learned preferences. Rather than relying on the explicitly trained reward model, DPO (Rafailov et al., 2023) simplifies the learning process by leveraging the insight that the optimal policy can be expressed in closed form using relative log-likelihoods under $\pi_\theta$ and $\pi_{\text{ref}}$:

$$\min_{\pi_\theta} \mathbb{E}_{(x,q,y_w,y_r)\sim\mathcal{D}}\Big[-\log\sigma\Big(\alpha\log\frac{\pi_\theta(y_w\mid x,q)}{\pi_{\text{ref}}(y_w\mid x,q)}-\alpha\log\frac{\pi_\theta(y_r\mid x,q)}{\pi_{\text{ref}}(y_r\mid x,q)}\Big)\Big]. \quad (5)$$

This formulation enables direct policy optimization from preference pairs, aligning the output probabilities of MLLMs with human preferences.

## 3 METHOD

We propose a token-adaptive min-max strategy with perturbations on visual-agnostic tokens and a frequency-based regularizer for improved alignment. An overview is shown in Figure 3, and the detailed algorithm is illustrated in Appendix B.

### 3.1 MIN-MAX REFORMULATION OF DPO

To address the limitations of traditional DPO, we reformulate preference optimization as a *token-adaptive min-max game*. The inner maximization introduces controlled token-level perturbations $\varphi(\cdot)$ to induce input distribution shifts, while the outer minimization aligns the policy $\pi_\theta$ with preference signals. Formally, we define the min–max preference objective as:

$$\min_{\pi_\theta}\max_{\varphi\in\Phi(\mathcal{A})}\mathbb{E}_{(x,q,y_w,y_r)\sim\mathcal{D}}\big[\mathcal{L}_{\text{TARS}}\big(x,\varphi(q),y_w,y_r\big)\big], \quad (6)$$

where $\varphi$ is a token-level perturbation function constrained to visually agnostic tokens, *i.e.*, $\Phi(\mathcal{A}) = \{\varphi \mid \{i \mid \varphi(q_i)\neq q_i\}\subseteq\mathcal{A}(x,q)\}$. This min–max objective promotes preference alignment under distributional shifts, helping to mitigate spurious correlations and reduce hallucinated outputs.

### 3.2 MAXIMIZING WITH TOKEN PERTURBATIONS

As shown in Equation (5), DPO aligns models with preferred responses via log-likelihood ratios against a reference model. However, we observe that this formulation can encourage overfitting to superficial patterns such as frequent phrases, stylistic tokens, which we find reduce effective alignment with the visual context in multimodal settings (Setlur et al., 2024; Fu et al., 2025).

To counter this, we apply token-wise maximization to introduce distribution shifts and reduce overfitting to preference signals. Formally, we define:

$$\varphi(q)=\arg\max_{\varphi\in\Phi(\mathcal{A})}\text{Sim}(\varphi(q),q), \quad (7)$$

where $\Phi(\mathcal{A})$ denotes allowable perturbations constrained to $\mathcal{A}(x, q)$, and $\mathrm{Sim}(\varphi(q), q)$ measures token-level deviation. In practice, we approximate $\varphi^*(q)$ by applying token-level transformations:

$$\varphi(q) = \{\mathbb{I}[i \in \mathcal{A}(x, q)] \cdot \varphi(q_i) + \mathbb{I}[i \notin \mathcal{A}(x, q)] \cdot q_i\}_{i=1}^{|q|}, \tag{8}$$

where $\varphi(q_i)$ is constructed using masking or synonym substitution. This approximation simulates worst-case alignment uncertainty while preserving semantic integrity.

To preserve semantics, we restrict changes to visual-agnostic tokens with minimal impact on cross-modal alignment. We compute token-level visual relevance as the similarity between visual features $\mathcal{G}_v(x)$ and token embeddings $\mathcal{G}_t(q_i)$. We then identify a set $\mathcal{A}$ of $N_t$ visually agnostic tokens with the lowest cross-modal alignment scores:

$$\mathcal{A} = \mathrm{Top}_{N_t}\left(-\mathcal{G}_v(x)\mathcal{G}_t(q_i)^T\right), N_t = \left\lfloor \omega \cdot \Delta P^{-1} \right\rfloor + 1, \tag{9}$$

where $\lfloor \cdot \rfloor$ denotes the floor operation, $\omega$ is a scaling coefficient. $P = -\mathcal{G}_v(x)\mathcal{G}_t(q_i)^T$ is the similarity score matrix, $\Delta P = \max_j P_j - \max_{k \neq j} P_k$ quantifies the predictive uncertainty of MLLMs. We adapt $N_t$ inversely to this margin: confident predictions lead to fewer perturbations, while greater uncertainty induces broader variation.

## 3.3 Spectral Regularization for Token Alignment

While our method introduces token-level perturbations to simulate distribution shifts, the supervision derived from preference pairs $(y_w, y_r)$ is static. This discrepancy between adaptive input representations and fixed supervision may encourage the model to learn distribution-specific artifacts, especially under strong alignment constraints (Fu et al., 2025; Chowdhury et al., 2024).

In practice, semantic alignment does not require strict token-level correspondence. Enforcing fine-grained constraints may reintroduce spurious correlations that our min–max strategy aims to mitigate (Zhou et al., 2024; Tian et al., 2025). To address this, we propose frequency-domain alignment, where local token perturbations translate into smooth variations in spectral space. This approach ensures semantic consistency between perturbed and inputs without rigid token-wise matching.

Specifically, we extract hidden states for $(x, \varphi(q), y_w)$ and contrast them with $(x, q, y_w)$ and $(x, q, y_r)$ using the FFT (Cooley & Tukey, 1965). Formally, let $z \in \mathbb{R}^{L \times D}$ denote a sequence of hidden states, where $L$ is the token length. We compute the spectral representation as:

$$\mathcal{F}(z) = \left| \mathrm{Re}\left[ \sum_{l=0}^{L-1} z_l \cdot e^{-2\pi i k l / L} \right] \right|_2, \text{ for } k = 0, \ldots, L-1, \tag{10}$$

where the FFT is applied along the token axis and $|\cdot|_2$ computes the $\ell_2$ norm over real-valued frequency magnitudes. The resulting spectral preference loss is defined as:

$$\mathcal{L}_{\mathrm{freq}} = -\log \sigma\left(\beta \cdot \left[ \log \frac{\mathcal{F}(h_\theta(x, \varphi(q), y_w))}{\mathcal{F}(h_{\mathrm{ref}}(x, q, y_w))} - \log \frac{\mathcal{F}(h_\theta(x, \varphi(q), y_w))}{\mathcal{F}(h_{\mathrm{ref}}(x, q, y_r))} \right]\right), \tag{11}$$

where $h_\theta(\cdot)$ and $h_{\mathrm{ref}}(\cdot)$ are the hidden states of the policy and reference models. This objective follows the logic of DPO but extends alignment to the spectral domain, improving consistency in frequency-aware representations and reducing hallucinations from overfitting to fixed preferences.

## 3.4 Minimization Objective in TARS

We integrate the standard DPO loss with spectral regularization to yield the final TARS training objective. Given a perturbed input $\varphi(q)$ obtained from the inner maximization, and its original counterpart $q$, the overall loss is defined as:

$$\mathcal{L}_{\mathrm{TARS}}(x, q, \varphi(q), y_w, y_r) = \mathcal{L}_{\mathrm{DPO}}(x, \varphi(q), y_w, y_r) + \lambda \cdot \mathcal{L}_{\mathrm{freq}}(x, q, \varphi(q), y_w, y_r), \tag{12}$$

where $\lambda$ is a weighting coefficient that balances preference alignment and spectral consistency. This formulation encourages the model to preserve causal alignment with preference signals, thereby mitigating spurious correlation.

## 4 EXPERIMENTS

### 4.1 EXPERIMENT DETAILS

**Experiment Setups.** We evaluate our approach on the multimodal large language model LLaVA-v1.5 (Liu et al., 2023b) at both 7B and 13B scales, and on Muffin-13B Yu et al. (2023); Fu et al. (2025). All evaluations are performed with greedy decoding and a temperature of 0.

To enable fair comparison, we align our training configuration with the most data-efficient preference optimization baselines. Specifically, we randomly sample 4.8k instances from the RLHF-V-Dataset (Yu et al., 2024), consistent with OPA-DPO (Yang et al., 2025), and adopt the same training strategy as CHiP-DPO (Fu et al., 2025). All models are trained on eight NVIDIA A100 (80GB) GPUs. We set $\alpha = 1$ in Equation (5) and $\beta = 1$ in Equation (11) for preference optimization. We implement $\varphi(\cdot)$ using both token masking (Mask) and replacement (Replace) strategies in Equation (8), and set the perturbation constraint strength to $\omega = 0.1$ in the adversarial min-max formulation Equation (9). We use a frequency-domain loss weight of $\lambda = 0.1$ in Equation (12). Full implementation details are provided in Appendix A, and ablation studies are reported in Appendix C.

**Evaluation Benchmark.** We evaluate TARS across both generative and discriminative hallucination benchmarks to ensure that hallucination mitigation does not come at the cost of factual grounding. Our evaluation framework includes four established benchmarks:

**1) AMBER** Wang et al. (2023) (Generative): Fine-grained benchmark of hallucination evaluation. In line with prior works Yang et al. (2025); Fu et al. (2025), we evaluate only the generative subset using the official codebase. Metrics include CHAIR Rohrbach et al. (2018), object coverage (Cover), response-level hallucination rate (Hal-Rate), and alignment with human cognition (Cog).

**2) MMHal** (Sun et al., 2023) (Generative): A VQA benchmark with real-world scenarios, evaluated using GPT-4V feedback to measure overall scores and hallucination rates (Hal-Rate).

**3) OBJHal** (Yu et al., 2024) (Generative): A benchmark evaluating hallucinations in captioning. We report hallucination rates at the response level ($CR_s$) and object mention level ($CR_i$).

**4) POPE** (Li et al., 2023) (Discriminative): A binary VQA benchmark designed to assess object hallucination along the textual axis through yes/no questions.

**Baseline Methods.** We compare against two categories:

**(1) Advanced multimodal foundation models:** Intern-VL2.5-7B (Chen et al., 2024e), Qwen-VL2.5-8B (Bai et al., 2025), DeepSeek-VL2-27B (Wu et al., 2024), and GPT-4o (Hurst et al., 2024).

**(2) LLaVA-v1.5 with RL techniques:** We evaluate multiple RL-based approaches applied to both the 7B and 13B variants of LLaVA-v1.5, including RLHF (Sun et al., 2024), RLAIF (Yu et al., 2025), HALVA (Sarkar et al., 2025a), as well as three state-of-the-art methods based on direct preference optimization (DPO): DPO (Pi et al., 2024), CHiP-DPO (Fu et al., 2025), and OPA-DPO (Yang et al., 2025). A comparison of algorithmic properties is provided in Table 3.

### 4.2 EVALUATION ON HALLUCINATION BENCHMARKS

Table 1 presents a performance comparison across four multimodal hallucination benchmarks. For TARS, we adopt two perturbation strategies: token masking and synonym replacement. TARS consistently reduces hallucinations under minimal supervision, scales effectively, preserves factual accuracy, and achieves competitive performance against proprietary models[1]. Our analysis highlights the following key findings:

**(1) Hallucination mitigation.** TARS achieves consistent reductions in hallucination across benchmarks. On the 7B scale, it lowers the AMBER hallucination rate from 35.4% to 13.2%, a 22.2 percentage point (pp) improvement. Concurrently, object coverage rises from 51.7% to 59.6% (+7.9 pp), and cognitive inconsistency (Cog) drops from 4.2 to 0.4 (–3.8 pp). On OBJHal, the response-level hallucination rate ($CR_s$) decreases from 54.0% to 12.0%.

---

[1]Extended results on Muffin are reported in Appendix E.

Table 1: Comparison across hallucination benchmarks. We evaluate SOTA MLLMs as reference baselines, denoted by §. For algorithms with available checkpoints, results from re-testing are marked with †; for those without, we reproduce results using settings from (Fu et al., 2025; Li et al., 2024), denoted by ‡. All experiments use greedy decoding with temperature set to 0 for consistency and reproducibility. **Bold** denotes the best performance, and underlined denotes the second-best.

| Algorithm | AMBER | | | | MMHal | | POPE | | OBJHal | |
|---|---|---|---|---|---|---|---|---|---|---|
| | CHAIR↓ | Cover↑ | Hal-Rate↓ | Cog↓ | Score↑ | Hal-Rate↓ | Acc↑ | Pre↑ | CR$_s$↓ | CR$_i$↓ |
| Intern-VL2.5-7B (Chen et al., 2024e)§ | 7.9 | 54.7 | 37.1 | 3.2 | 3.54 | 0.26 | - | - | 36.0 | 9.1 |
| Qwen-VL2.5-8B (Bai et al., 2025)§ | 4.6 | 54.6 | 21.1 | 1.3 | 3.29 | 0.27 | - | - | 40.7 | 8.6 |
| DeepSeek-VL2-27B (Wu et al., 2024)§ | 2.4 | 56.6 | 16.3 | 0.9 | 2.84 | 0.27 | - | - | 10.0 | 7.0 |
| GPT-4o (Hurst et al., 2024)§ | 2.5 | 60.9 | 17.6 | 0.8 | 3.87 | 0.24 | - | - | 29.3 | 6.7 |
| **LLaVA-v1.5-7B (Liu et al., 2023b)§** | 7.6 | 51.7 | 35.4 | 4.2 | 2.02 | 0.61 | 80.0 | 61.8 | 54.0 | 15.8 |
| + *RLHF* (Sun et al., 2024)† | 8.3 | 52.2 | 41.8 | 4.5 | 1.93 | 0.67 | 82.0 | 69.3 | 56.0 | 15.2 |
| + *RLAIF* (Yu et al., 2025)† | 3.0 | 50.3 | 16.5 | 1.0 | **2.89** | **0.42** | 88.1 | 88.0 | 13.7 | 4.2 |
| + *HALVA* (Sarkar et al., 2025a)† | 6.9 | 52.8 | 33.2 | 3.5 | 2.12 | 0.59 | 87.5 | 79.6 | 47.3 | 14.6 |
| + *DPO* (Li et al., 2024)‡ | 4.9 | 56.6 | 26.4 | 2.5 | 2.19 | 0.61 | 87.8 | 82.0 | 14.0 | 5.0 |
| + *CHiP-DPO* (Fu et al., 2025)‡ | 2.9 | 57.3 | 19.9 | 1.0 | 2.32 | 0.57 | 81.1 | 91.8 | **7.3** | 4.3 |
| + *OPA-DPO* (Yang et al., 2025)† | 2.7 | 47.4 | **12.5** | 0.9 | 2.78 | 0.46 | 87.4 | 86.2 | 13.3 | 4.5 |
| + TARS (Mask) | 2.4 | **59.6** | 13.2 | **0.4** | 2.48 | 0.45 | **88.7** | **97.5** | 12.0 | **3.2** |
| + TARS (Replace) | **2.1** | 59.3 | 14.9 | 0.7 | 2.54 | 0.46 | 87.9 | 97.0 | 13.4 | 3.3 |
| **LLaVA-v1.5-13B (Liu et al., 2023b)§** | 6.7 | 52.1 | 32.5 | 3.5 | 2.39 | 0.53 | 74.6 | 55.2 | 50.0 | 14.5 |
| + *RLHF* (Sun et al., 2024)† | 7.1 | 51.4 | 36.3 | 3.6 | 2.10 | 0.67 | 83.6 | 71.2 | 46.7 | 11.6 |
| + *HALVA* (Sarkar et al., 2025a)† | 6.5 | 53.4 | 30.1 | 3.3 | 2.28 | 0.56 | 86.8 | 75.6 | 42.7 | 12.1 |
| + *DPO* (Li et al., 2024)‡ | 4.1 | 56.7 | 24.3 | 2.2 | 2.48 | 0.50 | 85.2 | 84.3 | 19.0 | 7.2 |
| + *CHiP-DPO* (Fu et al., 2025)‡ | 3.8 | 58.6 | 20.8 | 1.7 | 2.70 | 0.46 | 86.6 | 74.9 | 30.0 | 6.2 |
| + *OPA-DPO* (Yang et al., 2025)† | 2.8 | 48.4 | 13.5 | 1.0 | **3.02** | **0.40** | 87.2 | 80.7 | 18.3 | 5.1 |
| + TARS (Mask) | **2.1** | **59.8** | **12.5** | 0.6 | 2.89 | 0.45 | **87.6** | **93.0** | 14.6 | **2.8** |
| + TARS (Replace) | **2.1** | 59.4 | 13.6 | 0.7 | 2.63 | 0.47 | 86.9 | 92.5 | 14.9 | 3.4 |

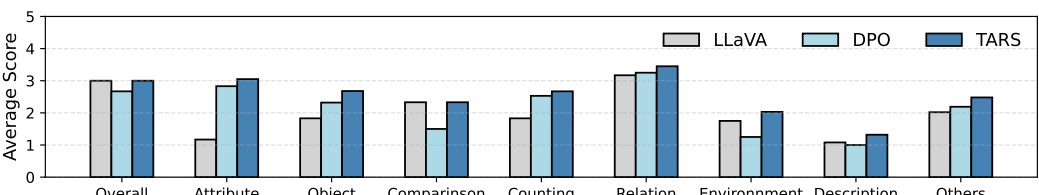

Figure 4: Comparison of average scores across question categories on the MMHal benchmark.

**(2) Data and supervision efficiency.** TARS achieves strong hallucination mitigation without relying on expert-labeled feedback or high-resource teacher models. As shown in Table 3, it uses only 4.8k public preference samples, matching the data budget of OPA-DPO, yet achieves superior performance on AMBER (13B) by improving object coverage from 48.4% to 59.8% (+11.4 pp) and reducing the hallucination rate from 13.5% to 12.5% (–1.0 pp).

**(3) Scalability.** TARS exhibits consistent performance as model capacity increases. From 7B to 13B, CHAIR improves from 2.4 to 2.1 (-0.3 pp) and hallucination rate drops from 13.2% to 12.5% (-0.7 pp). TARS-13B also surpasses all 13B baselines, confirming the scalability.

**(4) Factual consistency.** TARS effectively suppresses hallucinations without impairing factual understanding[2]. It achieves 88.7% accuracy on POPE (+8.7 pp over LLaVA-7B) in fine-grained visual reasoning and reduces object-level hallucination on OBJHal to 3.2% (–1.1 pp).

**(5) Competitiveness with proprietary models.** TARS delivers performance comparable to industrial-scale MLLMs. At 13B, it approaches GPT-4o in AMBER Cover (59.8% vs. 60.9%) and surpasses it in Hal-Rate (12.7% vs. 17.6%), while also outperforming DeepSeek-VL2-27B.

### 4.3 ABLATION ANALYSES ON COMPONENT

We analyze the key TARS components through ablations in Table 2, focusing on three elements:

**(1) Token-level perturbation (TP)** in Equation (6), which introduces distributional shifts and proves essential for revealing token-level vulnerabilities, its removal increases Cog from 0.4 to 2.5.

---

[2]Detailed generative, discriminative, and fine-grained analyses are provided in Appendix D.

Table 2: Ablation results of token-level perturbation (TP), cross-modal alignment score (CAS), and spectral preference alignment (SPA) under 7B scale.

| Algorithm | AMBER | | | OBJHal | |
|---|---|---|---|---|---|
| | Cover↑ | Hal-Rate↓ | Cog↓ | CR$_s$↓ | CR$_i$↓ |
| LLaVA (Liu et al., 2023b) | 51.7 | 35.4 | 4.2 | 54.0 | 15.8 |
| **TARS** | **59.6** | **13.2** | **0.4** | **12.0** | **3.2** |
| w/o **TP** | 56.6 | 26.4 | 2.5 | 14.0 | 5.0 |
| w/o **CAS** | 55.9 | 17.7 | 1.3 | 12.7 | 3.5 |
| w/o **SPA** | 58.3 | 15.1 | 0.7 | 12.5 | 3.7 |
| w/o **CAS&SPA** | 55.1 | 18.5 | 1.5 | 12.6 | 3.8 |

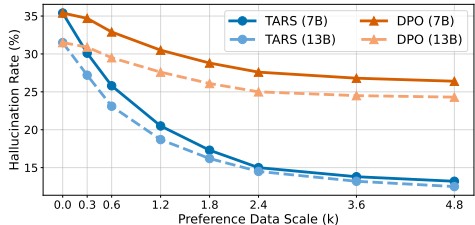

Figure 5: Comparison of AMBER hallucination rate versus preference data scale.

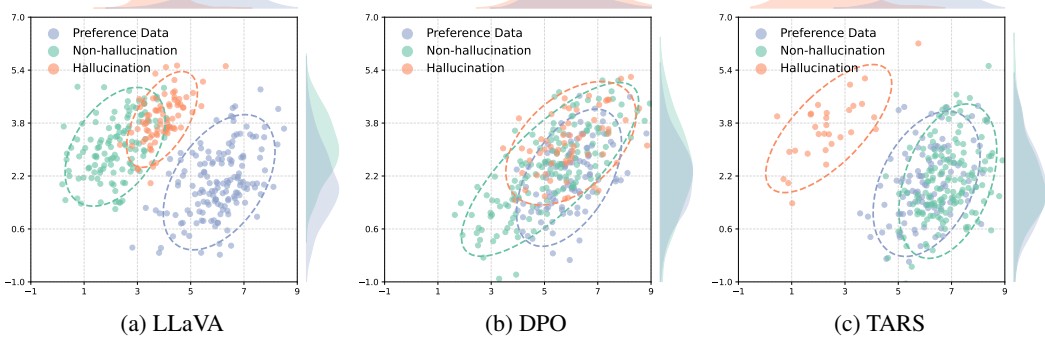

(a) LLaVA        (b) DPO        (c) TARS

Figure 6: Distribution of hidden representations across preference-aligned, non-hallucinated, and hallucinated responses of different MLLMs. Top and right margins show marginal distributions along key feature dimensions. We extract representations from 100 preference training instances and 200 AMBER inputs across text and vision modalities. Responses to AMBER inputs are categorized as non-hallucinated or hallucinated based on factual coherence. TARS aligns with preference data while avoiding overfitting to spurious correlations, demonstrating superior factual fidelity.

**(2) Cross-modal alignment score (CAS)** in Equation (9), which targets visually agnostic tokens to preserve semantic fidelity—its absence leads to a 4.5-point increase in hallucination and a 0.9 rise in Cog, indicating weaker suppression of spurious correlations.

**(3) Spectral preference alignment (SPA)** in Equation (11), which regularizes frequency-aware consistency—its removal increases the hallucination rate by 1.9 points and CR$_i$ from 3.2 to 3.7, suggesting degraded fine-grained factual grounding.

### 4.4 ABLATION ANALYSES ON PREFERENCE SCALE IMPACT

We investigate how the scale of preference data affects hallucination suppression in TARS by training on subsets of the 4.8k dataset and comparing with standard DPO (Figure 5). TARS consistently achieves lower hallucination rates across all scales and shows sharper improvements in early stages. From 0 to 1.8k examples, the 7B and 13B variants reduce hallucinations by over 18 and 15 percentage points, respectively. While the gains taper beyond 3.6k, performance remains stable, indicating strong data efficiency and effective use of limited supervision compared to DPO.

### 4.5 STABILITY OF SEMANTIC REPRESENTATIONS

We analyze how preference optimization reshapes hidden-state distributions in Figure 6.

**(1) Disentanglement of hallucinated and preference characteristics.** TARS yields a more structured latent space, where hallucinated and preference-aligned representations are separated. This separation indicates that hallucinations are not artifacts of spurious preference associations. In contrast, DPO exhibits entangled clusters, where hallucinated points are interwoven with preference representations, suggesting that DPO-trained models overfit superficial signals.

**(2) Selective alignment with non-hallucinated features.** TARS selectively aligns non-hallucinated responses with preference features while isolating hallucinated content in the representation space.

Table 3: Comparison of preference optimization strategies. TARS achieves hallucination mitigation and causal alignment with minimal data and no expert feedback.

| Algorithm | Data Size | Feedback | Reward-Free | Hallucination Mitigation | Causal Alignment |
|---|---|---|---|---|---|
| **LLaVA-v1.5** (Liu et al., 2023b) | - | - | ✗ | ✗ | ✗ |
| + *RLHF* (Sun et al., 2024) | 122k | self-reward | ✓ | ✗ | ✗ |
| + *RLAIF* (Yu et al., 2025) | 16k | LLaVA-Next | ✗ | ✓ | ✗ |
| + *HALVA* (Sarkar et al., 2025a) | 22k | GPT-4V | ✗ | ✗ | ✗ |
| + *DPO* (Li et al., 2024) | 5k | self-reward | ✓ | ✓ | ✗ |
| + *CHiP-DPO* (Fu et al., 2025) | 5k | self-reward | ✓ | ✓ | ✗ |
| + *OPA-DPO* (Yang et al., 2025) | 4.8k | GPT-4V | ✗ | ✓ | ✗ |
| + *TARS (Ours)* | 4.8k | self-reward | ✓ | ✓ | ✓ |

This alignment distinguishes TARS from both the scattered representations in base LLaVA and the feature entanglement in DPO models. Our findings show TARS creates a semantically faithful space by reinforcing only factually grounded responses that match learned preferences, avoiding amplification of spurious preference correlations.

## 5 RELATED WORK

Multimodal large language models (MLLMs) extend LLMs by integrating visual inputs to support multimodal reasoning (Chen et al., 2024b; Feng et al., 2025; Jain et al., 2024). Typically, visual features are extracted by a vision encoder, aligned through a connector, and processed by the LLM (Liu et al., 2023b; Parekh et al., 2024). Despite strong performance, MLLMs often produce factually incorrect or visually ungrounded outputs, undermining reliability (Bai et al., 2024; Chen et al., 2025). This issue is more severe than in unimodal LLMs (Chen et al., 2024d; Jiang et al., 2024a), mainly due to modality imbalance (Ma et al., 2024; He et al., 2024) and ineffective fusion (Bellagente et al., 2023; Ji et al., 2023). Recent studies attribute these failures to persistent misalignment between multimodal representations and human expectations, rather than model capacity (Chen et al., 2024c; Liu et al., 2024c; Ruan et al., 2025).

A key bottleneck in addressing MLLM hallucinations lies in aligning model outputs with human preferences for factual consistency. Unlike knowledge-intensive pretraining (Chang et al., 2024; McKinzie et al., 2024) and instruction tuning (Chen et al., 2024a; Liu et al., 2023b), recent methods typically leverage small-scale human preference data refined via reinforcement learning (Yu et al., 2024; Casper et al., 2023). Direct preference optimization (DPO)(Rafailov et al., 2023; Pi et al., 2024) has become a leading approach due to its simplicity and effectiveness, demonstrated in ChiP-DPO(Fu et al., 2025) and OPA-DPO (Yang et al., 2025). However, DPO's reliance on limited data can cause overfitting to superficial linguistic cues (Setlur et al., 2024; Fu et al., 2025), leading to distributional rigidity and reduced adaptability to modality-specific semantics (Ouali et al., 2024; Song et al., 2024). These limitations call for more adaptive alignment strategies that capture token-level variability and cross-modal dependencies.

To address these challenges, we propose a token-adaptive min-max alignment strategy that enhances preference learning without relying on high-resource expert feedback (e.g., GPT-4V). Using only a small public preference dataset, our method effectively mitigates hallucinations and consistently outperforms RL-based baselines across benchmarks. Table 3 compares preference optimization methods in terms of data scale, supervision, and alignment performance.

## 6 CONCLUSION

In this work, we introduce *TARS*, a novel lightweight strategy that reformulates direct preference optimization (DPO) as a min-max objective. TARS maximizes token-level distributional shifts under semantic constraints to simulate alignment uncertainty, while simultaneously minimizing the expected preference loss under these controlled perturbations. This formulation encourages the model to align more faithfully with causally grounded visual cues rather than overfit to superficial textual correlations, effectively mitigating hallucinations. TARS achieves strong hallucination suppression and consistently outperforms prior methods across most standard evaluation metrics, despite using only 4.8k public preference samples and no expert-labeled feedback or large-scale teacher models. Empirical results underscore the effectiveness of token-level alignment strategies for mitigating hallucinations in low-supervision settings.

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

## STATEMENT ABOUT THE USE OF LARGE LANGUAGE MODELS (LLMs)

We used large language models (LLMs) solely as a writing assist tool to check grammar and improve clarity of exposition. No part of the methodology, experiments, or analysis relied on LLMs.

## APPENDIX OVERVIEW

This appendix provides additional details to support the main paper. It is organized as follows:

- **Section A** details model configurations, training settings for DPO and TARS, and token perturbation procedures.
- **Section B** presents the min-max optimization algorithm of TARS in pseudocode form.
- **Section C** includes extended ablations analyzing the perturbation magnitude and spectral regularization strength.
- **Section D** reports additional benchmark results and fine-grained hallucination metrics.
- **Section E** presents results on Muffin-13B, demonstrating the generality of TARS across different MLLM architectures.
- **Section F** discusses model behavior, including sensitivity and design insights.
- **Section G** showcases qualitative comparisons on representative examples.

## A    IMPLEMENTATION DETAILS

### A.1    BASE MODEL SETUPS

We evaluate our method on LLaVA-v1.5 (Liu et al., 2023b) models with 7B and 13B parameters. LLaVA-v1.5 adopts Vicuna-7B/13B (Chiang et al., 2023) as the language backbone and CLIP-ViT-L/14 (Radford et al., 2021) as the vision encoder. The vision encoder also serves as the similarity function $\mathcal{G}(\cdot)$ used in Eq. (9) to compute alignment between visual inputs and text tokens. All experiments are conducted using greedy decoding with a temperature of 0 to ensure deterministic outputs and reproducibility.

### A.2    DPO TRAINING SETUPS

For fair comparison, DPO (Wang et al., 2024a), CHiP (Fu et al., 2025), and TARS follow the same training protocol as described in CHiP (Fu et al., 2025). Specifically, we set the number of epochs to 3, learning rate to 5e-7, warmup ratio to 0.03, maximum sequence length to 2048, and gradient clipping threshold to 20.0. Notably, TARS requires no task-specific hyperparameter tuning and demonstrates generalization across different base models and datasets. All experiments are conducted on 8×A100 GPUs (80GB). Each training run takes approximately 3.0 hours on LLaVA-v1.5-7B and 3.4 hours on LLaVA-v1.5-13B.

To generate perturbed inputs, we apply two token-level adversarial strategies: *replace* and *mask*. Both are guided by token similarity scores that estimate the alignment between each text token and the visual context. The similarity matrix is normalized into perturbation scores, such that tokens with lower alignment are more likely to be modified. In `replace` mode, these tokens are substituted with random vocabulary tokens. In `mask` mode, they are replaced with a special token such as `[MASK]`, `[UNK]`, or `[PAD]`, depending on tokenizer availability. Special tokens (e.g., `[BOS]`, `[EOS]`, `[PAD]`) are explicitly excluded from perturbation.

### A.3    EVALUATION BENCHMARK SETUPS

We follow the original evaluation settings and benchmark splits for AMBER (Wang et al., 2023), MMHal (Sun et al., 2023), and OBJHal (Yu et al., 2024) as specified in their respective papers. For POPE (Li et al., 2023), we construct a new benchmark of 9,000 VQA pairs by sampling using the `popular`, `random`, and `adversarial` strategies.

For evaluation metrics, we adopt four response-level hallucination measures across different benchmarks: CHAIR (Rohrbach et al., 2018) for object hallucination detection, object coverage (Cover) for completeness measurement, response-level hallucination rate (Hal-Rate) for overall hallucination assessment, sentence-level hallucination rate ($CR_s$) for holistic response evaluation, and object mention-level hallucination rate ($CR_i$) for fine-grained object-level analysis.

For evaluation feedback collection, we employ the `en-core-web-lg` English NLP pipeline for AMBER to extract structured semantic cues as lightweight and reproducible evaluators. For MMHal and OBJHal, we utilize the expert GPT-4V model (Hurst et al., 2024) (`gpt-4-1106-vision-preview`) for feedback evaluation, following the established protocols.

---

**Algorithm 1** *TARS Training Procedure*

---

**Inputs:** Trainable policy $\pi_\theta$, reference policy $\pi_{\text{ref}}$, and preference dataset $\mathcal{D} = \{x, q, y_w, y_r\}^N$.
**Encoders:** Visual encoder $\mathcal{G}_v$; text encoder $\mathcal{G}_t$.
**Hyperparameters:** DPO scaling $\alpha$, perturbation ratio $\omega$, frequency scaling $\beta$, loss weight $\lambda$.

1:  **for** each epoch **do**
2:    Sample preference tuple $(x, q, y_w, y_r) \sim \mathcal{D}$.
3:    *Max Part:*
4:    Compute token-level visual relevance:

$$P_i = \mathcal{G}_v(x) \cdot \mathcal{G}_t(q_i)^T. \tag{13}$$

5:    Estimate model confidence margin:

$$\Delta P = \max_j P_j - \max_{k \neq j} P_k. \tag{14}$$

6:    Determine adaptive perturbation budget:

$$N_t = \lfloor \omega \cdot \Delta P^{-1} \rfloor + 1. \tag{15}$$

7:    Select visually agnostic tokens:

$$\mathcal{A} = \text{Top}_{N_t}(-P). \tag{16}$$

8:    Apply controlled perturbation to obtain $\varphi(q)$:

$$\varphi(q) = \{\mathbb{I}[i \in \mathcal{A}(x, q)] \cdot \varphi(q_i) + \mathbb{I}[i \notin \mathcal{A}(x, q)] \cdot q_i\}_{i=1}^{|q|}. \tag{17}$$

9:    *Min Part:*
10:   Compute the preference alignment loss via DPO:

$$\mathcal{L}_{\text{DPO}} = -\log \sigma \left( \alpha \log \frac{\pi_\theta(y_w \mid x, \varphi(q))}{\pi_{\text{ref}}(y_w \mid x, \varphi(q))} - \alpha \log \frac{\pi_\theta(y_r \mid x, \varphi(q))}{\pi_{\text{ref}}(y_r \mid x, \varphi(q))} \right). \tag{18}$$

11:   Apply frequency-domain regularization:

$$\mathcal{L}_{\text{freq}} = -\log \sigma \left( \beta \cdot \left[ \log \frac{\mathcal{F}\left(h_\theta(x, \varphi(q), y_w)\right)}{\mathcal{F}\left(h_{\text{ref}}(x, q, y_w)\right)} - \log \frac{\mathcal{F}\left(h_\theta(x, \varphi(q), y_w)\right)}{\mathcal{F}\left(h_{\text{ref}}(x, q, y_r)\right)} \right] \right). \tag{19}$$

12:   Compute final objective:

$$\mathcal{L}_{\text{TARS}} = \mathcal{L}_{\text{DPO}} + \lambda \cdot \mathcal{L}_{\text{freq}}. \tag{20}$$

13:   Update $\pi_\theta$ via gradient descent.
14: **end for**
**Learned Policy:** Optimized policy $\pi_\theta^*$.

---

Table 4: Ablation study on the effect of token-level perturbation magnitude $\omega$ in TARS. We evaluate how varying the perturbation ratio influences hallucination suppression, semantic coherence, and grounding performance across four benchmarks. All experiments use LLaVA-v1.5-13B as the base model and adopt greedy decoding with temperature set to 0 for consistency. Bold results indicate the best-performing configuration.

| Perturbation Magnitude | AMBER | | | | MMHal | | POPE | | OBJHal | |
|---|---|---|---|---|---|---|---|---|---|---|
| | CHAIR↓ | Cover↑ | Hal-Rate↓ | Cog↓ | Score↑ | Hal-Rate↓ | Acc↑ | Pre↑ | $CR_s$↓ | $CR_i$↓ |
| **Referenced Results** | | | | | | | | | | |
| LLaVA-v1.5-7B | 7.6 | 51.7 | 35.4 | 4.2 | 2.02 | 0.61 | 80.0 | 61.8 | 54.0 | 15.8 |
| **TARS (Ours)** | | | | | | | | | | |
| $\omega = 1e{-}4$ | 3.3 | 58.0 | 15.4 | 1.3 | 2.35 | 0.50 | 86.0 | 95.0 | 16.2 | 4.6 |
| $\omega = 3e{-}4$ | 3.0 | 58.3 | 15.1 | 1.1 | 2.38 | 0.50 | 86.9 | 95.4 | 15.4 | 4.3 |
| $\omega = 5e{-}4$ | 2.9 | 58.7 | 14.5 | 0.8 | 2.41 | 0.47 | 87.7 | 96.1 | 13.6 | 3.8 |
| $\omega = 1e{-}3$ | **2.4** | **59.6** | **13.2** | **0.4** | **2.48** | **0.45** | **88.7** | **97.5** | **12.0** | **3.2** |
| $\omega = 5e{-}3$ | 3.3 | 57.8 | 15.9 | 1.4 | 2.29 | 0.48 | 86.9 | 91.2 | 16.1 | 3.9 |
| $\omega = 1e{-}2$ | 4.0 | 56.9 | 20.2 | 1.9 | 2.23 | 0.51 | 84.0 | 85.7 | 21.9 | 5.6 |

# B    ALGORITHM FLOWCHART

We present the full training procedure of TARS in Algorithm 1, which explicitly decomposes the learning process into two stages: a maximization phase that generates token-level perturbations based on visual relevance (*Max Part*), and a minimization phase that optimizes the model with preference supervision (*Min Part*). This min-max formulation allows TARS to effectively regularize overconfident preference patterns by injecting controlled distributional shifts during training. The maximization step identifies visually agnostic tokens and perturbs them via masking or replacement, while the minimization step jointly optimizes a DPO loss and a frequency-domain alignment objective. Overall, TARS effectively suppresses spurious token-visual correlations and significantly reduces hallucinations in multimodal preference optimization.

# C    EXTENDED ABLATION STUDIES

## C.1    IMPACT OF TOKEN-LEVEL PERTURBATION MAGNITUDE

We vary the token-level perturbation ratio $\omega$ and report results in Table 4 to investigate how perturbation strength affects model performance. In our method, Equation (9) governs the selection of tokens for perturbation based on their visual irrelevance. Specifically, we compute the similarity between visual features $\mathcal{G}_v(x)$ and text token embeddings $\mathcal{G}_t(q_i)$ to estimate token-level visual alignment. Tokens with the lowest scores are considered visual-agnostic and thus are most eligible for perturbation. The perturbation budget $N_t$ is adaptively determined via the scaling coefficient $\omega$ and model uncertainty $\Delta P$ as:

$$\mathcal{A} = \text{Top}_{N_t}(-\mathcal{G}_v(x) \cdot \mathcal{G}t(q_i)^T), \quad N_t = \lfloor \omega \cdot \Delta P^{-1} \rfloor + 1, \tag{21}$$

where $\Delta P = \max_j P_j - \max k \neq j P_k$ quantifies the margin between the top two token-level alignment scores. This formulation encourages stronger perturbations under high uncertainty and milder changes when the model is confident.

As shown in Table 4, moderate values of $\omega$ lead to optimal hallucination suppression across both AMBER and OBJHal. Excessively low or high perturbation strengths either under-regularize or destabilize training. When $\omega$ is too small (*e.g.*, $1e{-}4$), the induced distributional shift is limited, resulting in marginal improvement over the baseline. This insufficient perturbation fails to adequately expose the model's reliance on spurious token-level correlations, leading to suboptimal alignment correction. Conversely, overly large values (*e.g.*, $5e{-}3$ or $1e{-}2$) introduce excessive perturbation into visual regions, disrupting the semantic coherence of inputs. This degrades both hallucination control and downstream task accuracy, as the model overfits to unstable signals. The best results are obtained at $\omega = 1e{-}3$, which achieves a balance between perturbation diversity and input integrity.

## C.2  IMPACT OF FREQUENCY REGULARIZATION

To assess the contribution of spectral regularization, we conduct an ablation study by varying the frequency loss weight $\lambda$ in the TARS objective (Equation (12)). The term $\mathcal{L}_{\text{freq}}$ encourages semantic consistency between original and perturbed representations by aligning their frequency spectra. While conceptually appealing, the strength of this constraint is modulated by $\lambda$ and may interact non-trivially with preference supervision.

We evaluate $\lambda \in \{0.01, 0.02, 0.05, 0.10, 0.20, 0.50, 1.00\}$ and present results in Table 5. We omit $\lambda = 0.00$ here as it reduces the method to DPO with perturbation. As shown in the table, introducing spectral alignment leads to consistent improvements across all benchmarks. Performance improves steadily as $\lambda$ increases from 0.01 to 0.20, with hallucination rates (CHAIR, Hal-Rate, $CR_s$, $CR_i$) decreasing and coverage improving. The best trade-off is achieved at $\lambda = 0.20$, where TARS achieves the lowest hallucination and strongest grounding.

Beyond this point, we observe diminishing or adverse effects. For example, at $\lambda = 0.50$ and $\lambda = 1.00$, performance begins to degrade, particularly on MMHal and OBJHal. This trend suggests that overly aggressive regularization may constrain the model's ability to accommodate subtle semantic variations introduced by token-level perturbations, leading to underfitting or conservative outputs.

These results confirm that spectral alignment is an effective regularizer when applied with moderate strength. It improves semantic coherence across perturbed samples without rigidly enforcing token-level correspondence, thus allowing preference optimization to remain robust yet expressive under controlled distributional shifts.

Table 5: Ablation study on the effect of spectral alignment weight $\lambda$. We evaluate the impact of varying $\lambda$ on hallucination suppression and multimodal alignment. All experiments are conducted using LLaVA-v1.5-13B as the base model and employ greedy decoding with temperature set to 0 for consistency. Bold numbers indicate the best across each metric.

| Spectral Coefficient | AMBER | | | | MMHal | | POPE | | OBJHal | |
|---|---|---|---|---|---|---|---|---|---|---|
| | CHAIR↓ | Cover↑ | Hal-Rate↓ | Cog↓ | Score↑ | Hal-Rate↓ | Acc↑ | Pre↑ | $CR_s$↓ | $CR_i$↓ |
| **Referenced Results** | | | | | | | | | | |
| LLaVA-v1.5-13B | 6.7 | 52.1 | 32.5 | 3.5 | 2.39 | 0.53 | 74.6 | 55.2 | 50.0 | 14.5 |
| **TARS (Ours)** | | | | | | | | | | |
| $\lambda = 0.01$ | 2.9 | 58.7 | 14.8 | 1.0 | 2.80 | 0.48 | 87.2 | 92.5 | 15.4 | 3.7 |
| $\lambda = 0.02$ | 2.7 | 59.0 | 14.1 | 0.8 | 2.83 | 0.46 | 87.5 | 93.2 | 14.8 | 3.4 |
| $\lambda = 0.05$ | 2.6 | 59.3 | 13.5 | 0.6 | 2.85 | 0.46 | 87.6 | 93.5 | 14.7 | 3.1 |
| $\lambda = 0.10$ | 2.4 | 59.6 | 13.2 | **0.4** | 2.88 | **0.45** | 88.2 | 94.3 | 13.2 | **2.9** |
| $\lambda = 0.20$ | **2.1** | **59.8** | **12.5** | 0.6 | **2.89** | **0.45** | **88.5** | **95.0** | **12.8** | 2.8 |
| $\lambda = 0.50$ | 2.6 | 59.0 | 13.9 | 0.9 | 2.86 | 0.46 | 87.8 | 92.4 | 14.4 | 3.5 |
| $\lambda = 1.00$ | 3.0 | 58.2 | 15.1 | 1.3 | 2.81 | 0.47 | 86.7 | 91.0 | 15.6 | 4.1 |

# D  ADDITIONAL EXPERIMENTAL RESULTS

We present extended results on the AMBER benchmark in Table 6, evaluating hallucination performance from both generative and discriminative perspectives.

The left portion of the table reports generative metrics, including CHAIR, Coverage, Hallucination Rate, and Cognitive Score. TARS achieves substantial improvements across all, reducing hallucination by over 13 points compared to DPO, and significantly improving image grounding as reflected by higher coverage and cognitive consistency.

Beyond generative evaluation, we further introduce fine-grained discriminative metrics that assess hallucination across four categories: Existence, Relation, Attribute, and Action. This allows a more detailed understanding of where hallucinations occur and how well each method suppresses them. As shown in the right half of the table, TARS consistently outperforms both DPO and the LLaVA baseline in all dimensions. Notably, it excels in Relation and Attribute grounding, where conventional methods often struggle due to subtle cross-modal mismatches.

Together, these results underscore the strength of our token-adaptive perturbation strategy, which not only reduces hallucinations at a global level but also enhances semantic fidelity in specific visual grounding aspects, without relying on hand-crafted heuristics or additional supervision.

To further dissect model performance on hallucination-prone scenarios, we report average scores across different question categories in the MMHal benchmark (Figure 4). TARS consistently achieves higher scores across all categories, particularly excelling in fine-grained tasks involving spatial reasoning and attribute identification. These results suggest that our method improves not only overall hallucination rates but also robustness to diverse multimodal challenges, highlighting its effectiveness in aligning responses with nuanced visual semantics.

Table 6: Comparison of generative and fine-grained discriminative hallucination metrics on the AMBER benchmark. TARS achieves consistent gains over DPO and the LLaVA baseline across both holistic and category-specific evaluations, demonstrating enhanced visual-textual alignment and robust hallucination suppression.

| Algorithm | Generative | | | | Discriminative | | | |
|---|---|---|---|---|---|---|---|---|
| | CHAIR↓ | Cover↑ | Hal-Rate↓ | Cog↓ | Existence↑ | Relation↑ | Attribute↑ | Action↑ |
| **LLaVA-v1.5-7B** (Liu et al., 2023b) | 7.9 | 54.7 | 37.1 | 3.2 | 82.9 | 58.6 | 65.6 | 70.1 |
| + *DPO* (Li et al., 2024) | 4.9 | 56.6 | 26.4 | 2.5 | 87.1 | 59.7 | 74.6 | 79.4 |
| + *TARS (Ours)* | **2.4** | **59.6** | **13.2** | **0.4** | **95.3** | **62.8** | **78.6** | **86.5** |
| **LLaVA-v1.5-13B** (Liu et al., 2023b) | 6.7 | 52.1 | 32.5 | 3.5 | 94.1 | 45.5 | 70.1 | 76.2 |
| + *DPO* (Li et al., 2024) | 4.1 | 56.7 | 54.3 | 2.2 | 95.0 | 58.8 | 73.1 | 81.5 |
| + *TARS (Ours)* | **2.1** | **59.8** | **12.5** | **0.6** | **98.9** | **67.0** | **82.0** | **86.6** |

# E    ADDITIONAL RESULTS ON MUFFIN

We further validate TARS on the Muffin-13B architecture (Table 7). Consistent with our findings on LLaVA, both perturbation strategies yield substantial improvements over DPO and CHiP-DPO. TARS with token masking achieves the strongest overall performance, while synonym replacement remains competitive as the second-best variant.

These results confirm the versatility of our approach: TARS not only mitigates hallucinations more effectively than prior alignment methods but also preserves factual accuracy and improves coverage across diverse benchmarks. Importantly, the consistent gains across two distinct MLLM backbones underscore the generality of our token-adaptive perturbation framework, highlighting its potential as a plug-and-play strategy for robust multimodal alignment.

Table 7: Comparison of hallucination benchmarks on alternative MLLM architectures. We evaluate SOTA MLLMs as reference baselines, denoted by §. For algorithms without available checkpoints, we reproduce results using settings from (Fu et al., 2025), denoted by ‡. All experiments use greedy decoding with temperature set to 0 for consistency and reproducibility. **Bold** denotes the best performance, and underlined denotes the second-best.

| Algorithm | AMBER | | | | MMHal | | POPE | | OBJHal | |
|---|---|---|---|---|---|---|---|---|---|---|
| | CHAIR↓ | Cover↑ | Hal-Rate↓ | Cog↓ | Score↑ | Hal-Rate↓ | Acc↑ | Pre↑ | $CR_s$↓ | $CR_i$↓ |
| Intern-VL2.5-7B (Chen et al., 2024e)§ | 7.9 | 54.7 | 37.1 | 3.2 | 3.54 | 0.26 | - | - | 36.0 | 9.1 |
| Qwen-VL2.5-8B (Bai et al., 2025)§ | 4.6 | 54.6 | 21.1 | 1.3 | 3.29 | 0.27 | - | - | 40.7 | 8.6 |
| DeepSeek-VL2-27B (Wu et al., 2024)§ | 2.4 | 56.6 | 16.3 | 0.9 | 2.84 | 0.27 | - | - | 10.0 | 7.0 |
| GPT-4o (Hurst et al., 2024)§ | 2.5 | 60.9 | 17.6 | 0.8 | 3.87 | 0.24 | - | - | 29.3 | 6.7 |
| **Muffin-13B** (Yu et al., 2023)§ | 7.5 | 45.7 | 34.6 | 3.4 | 2.27 | 0.58 | 83.0 | 80.7 | 47.3 | 15.2 |
| + *RLHF* (Sun et al., 2024)‡ | 7.1 | 45.2 | 37.1 | 3.5 | 2.12 | 0.64 | 84.0 | 79.8 | 45.5 | 12.7 |
| + *DPO* (Li et al., 2024)‡ | 6.0 | 46.4 | 29.6 | 2.8 | 2.45 | 0.55 | 83.7 | 81.2 | 43.8 | 13.9 |
| + *CHiP-DPO* (Fu et al., 2025)‡ | 4.8 | 48.2 | 18.9 | 1.7 | 2.70 | 0.47 | 84.5 | 82.1 | 35.2 | 11.5 |
| + TARS (Mask) | **3.6** | **49.5** | **16.2** | **1.4** | 2.75 | **0.41** | **87.4** | **84.9** | **28.7** | **8.2** |
| + TARS (Replace) | 4.0 | 48.9 | 16.5 | 1.5 | **2.76** | 0.43 | 86.5 | 83.8 | 29.3 | 8.8 |

# F    DISCUSSIONS AND INSIGHTS

## F.1    WHY TARS OUTPERFORMS DPO: BEYOND NUMBERS

While TARS consistently outperforms standard DPO across hallucination benchmarks, its effectiveness stems not only from empirical gains, but from the design principles that enable better preference

alignment under uncertainty. Below, we outline the key factors contributing to the superior performance of TARS.

**Token-level perturbation enhances alignment robustness.** DPO relies on static textual inputs, making it susceptible to overfitting on superficial linguistic patterns such as high-frequency phrases or stylistic biases present in preference data. TARS addresses this issue by introducing controlled perturbations on visually agnostic tokens. These perturbations simulate semantically equivalent variations, thereby exposing the model to distributional shifts during training. As a result, the learned policy becomes more robust to alignment uncertainty and is encouraged to rely on visual grounding cues rather than memorized textual artifacts.

**Visual-agnostic targeting preserves grounding fidelity.** Unlike random or uniform perturbation strategies, TARS selectively perturbs tokens with low cross-modal relevance, those that carry minimal visual grounding. This design ensures that semantic shifts are injected without disrupting the causal connection between image and text. By isolating visually agnostic components for perturbation, TARS avoids damaging critical multimodal alignments, resulting in faithful responses that remain sensitive to visual semantics while being resilient to linguistic noise.

**Spectral alignment encourages semantic consistency.** To bridge the mismatch between input perturbations and static supervision, TARS introduces a spectral regularizer that aligns representations in the frequency domain. This global constraint allows for flexible modifications while maintaining semantic coherence at the sequence level. Unlike rigid token-level matching, frequency alignment smooths over local variations and discourages the model from latching onto spurious token-level correlations. This helps prevent overfitting to distribution-specific artifacts and improves generalization under preference supervision.

Table 8: Comparison of spectrum-based alignment (TARS) versus token-level contrastive alignment. Both models use identical perturbation policies. Spectrum-based alignment achieves lower hallucination and better semantic consistency.

| Alignment Strategy | AMBER | | | | MMHal | | POPE | | OBJHal | |
|---|---|---|---|---|---|---|---|---|---|---|
| | CHAIR↓ | Cover↑ | Hal-Rate↓ | Cog↓ | Score↑ | Hal-Rate↓ | Acc↑ | Pre↑ | $CR_s$↓ | $CR_i$↓ |
| Token-level (contrastive) | 3.4 | 57.2 | 16.3 | 1.4 | 2.36 | 0.49 | 87.1 | 93.3 | 15.8 | 4.9 |
| Spectrum-based (TARS) | **2.4** | **59.6** | **13.2** | **0.4** | **2.48** | **0.45** | **88.7** | **97.5** | **12.0** | **3.2** |

## F.2 TOKEN PERTURBATION SENSITIVITY

**Impact of perturbation strength.** TARS determines the number of perturbed tokens based on the model's confidence and a scaling factor $\omega$. In Appendix C.1, we conduct a systematic evaluation of perturbation magnitude to assess sensitivity. Results show that moderate perturbation levels (e.g., $\omega = 1e-3$) yield optimal hallucination suppression, while excessively small or large values lead to under- or over-regularization. These findings confirm that the model benefits from controlled perturbation, and that TARS is robust across a range of $\omega$ values when properly calibrated.

**Stability across perturbation strategies.** We also compare two token-level perturbation strategies used in TARS: token masking and synonym replacement. Despite their distinct mechanisms—masking introduces structural noise, while replacement maintains fluent semantics—both consistently outperform unperturbed DPO. Notably, masking yields slightly better hallucination mitigation, likely due to its stronger distributional shift. The close performance of both strategies suggests that the core effectiveness of TARS is not overly sensitive to the specific perturbation operator, as long as semantic integrity is preserved.

**Effectiveness of adaptive perturbation scope.** Instead of applying a fixed number of perturbations, TARS dynamically scales the perturbation budget according to the model's alignment uncertainty. This adaptive strategy ensures that confident predictions remain mostly intact, while uncertain ones are regularized more aggressively. Such input-aware perturbation improves training stability and avoids unnecessary semantic drift, reinforcing the model's ability to distinguish visually grounded content from spurious correlations.

### F.3 SPECTRUM-BASED ALIGNMENT VS TOKEN-LEVEL ALIGNMENT

**Motivation.** While TARS perturbs visually agnostic tokens to expose latent alignment vulnerabilities, the preference supervision signal remains static. This creates a mismatch: dynamic inputs versus fixed feedback. A common solution is to enforce token-level representation consistency, such as applying cosine similarity or contrastive loss between the perturbed and original hidden states. However, such approaches assume strict token-wise correspondence, which may not hold under semantic-preserving perturbations.

**Limitations of token-level alignment.** We empirically observe that enforcing rigid token-level matching often leads to instability during training and degrades hallucination mitigation. Token-level losses tend to penalize even small, semantically irrelevant shifts introduced by benign perturbations. This over-constraint reintroduces the spurious correlation that TARS aims to alleviate. In contrast, frequency-based alignment allows for local flexibility while enforcing global consistency in the hidden representation spectrum.

**Comparative evaluation.** To validate this claim, we compare the spectral preference loss in Equation (11) with a baseline token-level contrastive alignment strategy:

- **Contrastive loss (Token-level):** Hidden states from $(x, q, y_w)$ and $(x, \phi(q), y_w)$ are aligned using an InfoNCE objective.
- **Spectrum loss (TARS):** Frequency magnitude alignment via log-ratio of FFT-transformed hidden states.

As shown in Table 8, spectrum-based alignment achieves lower hallucination rates and better factual grounding across AMBER and OBJHal. This confirms that semantic-level flexibility, rather than rigid token consistency, is key to robust preference optimization.

### F.4 LIMITATIONS

In this work, we adopt two simple perturbation strategies: token masking and synonym replacement. These methods are chosen for their clarity, efficiency, and ease of interpretation, allowing us to isolate the effects of token-level alignment without introducing unnecessary complexity. However, their simplicity may limit the generality and flexibility of the approach. Future work could explore adaptive or data-driven perturbation mechanisms that better balance semantic preservation with distributional shift. Additionally, the current token selection strategy, based on cross-modal similarity heuristics, could be enhanced by learning-based relevance estimation or causal attribution techniques to enable more precise and effective perturbations.

## G QUALITATIVE EXAMPLES

We provide qualitative comparisons between standard DPO and our proposed TARS in Table 9, across diverse image-question pairs. TARS consistently demonstrates improved grounding and hallucination suppression, outperforming traditional DPO in several key aspects:

**Reduced hallucination via improved visual grounding.** Compared to DPO, TARS produces responses that more accurately reflect the image content. In all cases, DPO introduces visual details not present in the input, while TARS remains faithful to the scene.

**No degradation in response completeness.** Importantly, TARS maintains response richness without sacrificing length or informativeness. As illustrated in (d), TARS generates a detailed yet grounded answer, whereas DPO provides longer responses but includes hallucinated attributes.

**Better fine-grained grounding.** TARS exhibits improved alignment at the fine-grained level, such as object color and count. In examples (b) and (e), it correctly identifies color attributes that DPO misrepresents.

**Enhanced descriptive richness.** TARS responses incorporate more relevant visual details, indicating stronger cross-modal alignment. The generated answers are not only more accurate but also semantically richer, reflecting a deeper understanding of the visual input.

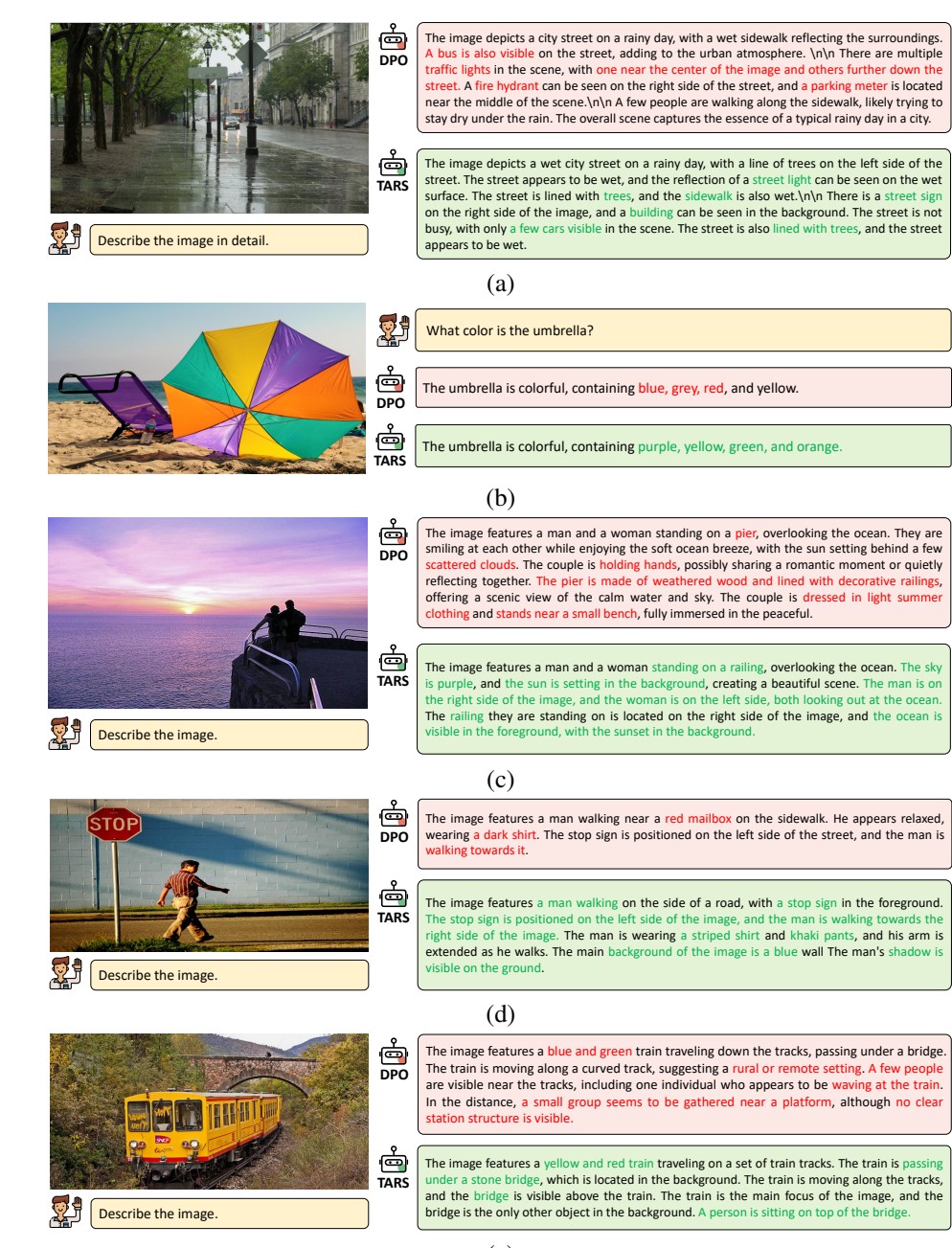

(a)

(b)

(c)

(d)

(e)

Table 9: Qualitative comparisons between DPO and our proposed TARS across five diverse image-prompt pairs, denoted as (a)–(e). Each row presents the same visual input and accompanying question, with model responses shown for both methods. Hallucinated content is highlighted in red, while accurate visual grounding is marked in green. TARS consistently produces more faithful and informative responses, demonstrating superior grounding and hallucination mitigation even under visually ambiguous or linguistically subtle conditions.

These examples highlight the effectiveness of token-adaptive perturbation in guiding the model to learn robust visual-textual associations, ultimately reducing hallucinations without compromising informativeness.

