# OpenReview forum: "TARS: MinMax Token-Adaptive Preference Strategy for MLLM Hallucination Reduction"
_ICLR.cc/2026/Conference — ICLR 2026 Conference Withdrawn Submission_

### Official Review · Reviewer_f4yi · 2025-10-30

**Soundness:** 2
**Presentation:** 3
**Contribution:** 2
**Rating:** 4
**Confidence:** 4

**Summary:**

This paper proposes TARS, a token-adaptive min-max strategy to mitigate hallucinations in MLLM. It reforms DPO into a min-max objective: 1) maximize token-level distributional shifts to simulate alignment uncertainty; 2) minimize preference loss with spectral regularization. Using only 4.8k preference samples, TARS outperforms DPO baselines on LLaVA-v1.5 on hallucinations.

**Strengths:**

1. The method is data-efficient as it only uses 4.8k records.
2. The overall performance on the hallucination reduction is sound and surpasses many other baselines.

**Weaknesses:**

1. The paper does not show the performance on the general MLLM/LLM benchmark, which makes me concerned that the performance over the hallucination is increased while the general performance is degrading.
2. In the paper, the authors suppose that the visual-text relevance could be computed from the similarity of the dot product between visual features and token embeddings. This seems to be intuitive, while it would help if some real examples were visualized to check whether the relevance could be calculated in this way.
3. The token-level perturbation seems to only ablate on the visually agnostic tokens. An ablation of the random token perturbation on the replace and mask strategy is needed.
4. The proposed frequency-domain alignment aims at ensuring semantic consistency, while no ablation on semantic similarity is done over the spectral regularization, only the hallucination metrics.
5. Though it may introduce more experiments and more computation, I still recommend that the author fully fine-tune the RLHF-V to see its scaling ability. A randomly sampled subset will introduce more randomness.

**Questions:**

N/A

---

### Official Review · Reviewer_Debk · 2025-10-31

**Soundness:** 2
**Presentation:** 2
**Contribution:** 2
**Rating:** 4
**Confidence:** 4

**Summary:**

This paper introduces TARS, a Token-Adaptive Preference Strategy designed to reduce hallucinations in Multimodal Large Language Models (MLLMs). The authors identify that standard Direct Preference Optimization (DPO) methods, which align model outputs with human preferences, often overfit to superficial textual cues in the preference data, leading to "distributional rigidity" and increased hallucinations. TARS reformulates DPO as a min-max optimization problem: it first *maximizes*token-level distributional shifts by perturbing "visual-agnostic" tokens (textual elements with minimal grounding in the image) to simulate uncertainty and break spurious correlations, and then *minimizes*the preference loss under these controlled perturbations. This approach encourages the model to rely on causally relevant visual information rather than textual patterns.

**Strengths:**

1. The paper claims standard DPO overfits to spurious text tokens and proposes TARS, which perturbs those “visual-agnostic tokens” to reduce hallucination.
2. TARS reports better grounding / lower hallucination on benchmarks like AMBER and MMHal, sometimes close to much larger models, using only ~4.8k preference pairs.

**Weaknesses:**

1. The method relies on computing a similarity score between each text token embedding and the visual features, then taking the tokens with lowest cross-modal similarity as “visual-agnostic,” and perturbing only those. This is a strong assumption: it assumes (i) that dot-product similarity between Gᵥ(x) and Gₜ(qᵢ) is a valid proxy for whether a token is visually grounded, and (ii) that low-similarity tokens are in fact safe to perturb without changing semantics. The paper does not rigorously justify this assumption with citations to prior work, nor does it provide its own empirical validation.
2. Randomly masking/replacing those tokens may break grammar and semantics, and they don’t show that general VQA / reasoning quality is not harmed. Recommend authors evaluate their method in MM-Vet, HalluBench, etc.
3. All main results are on older baselines like LLaVA-1.5/Muffin; they don’t apply TARS to stronger recent MLLMs such as Qwen 2.5 VL, so generality is unclear.
4. Equation (11) seems inconsistent with standard DPO: the second term’s numerator uses $y_w$ again, but it should likely be $y_r$; this needs correction.

**Questions:**

Please refer to the weaknesses part.

---

### Official Review · Reviewer_iypP · 2025-11-03

**Soundness:** 2
**Presentation:** 3
**Contribution:** 2
**Rating:** 4
**Confidence:** 4

**Summary:**

The paper proposes TARS, a preference-learning framework that reformulates DPO into a token-adaptive min–max method for multimodal LLMs. The inner maximization perturbs visual-agnostic text tokens to simulate controlled distribution shifts; the outer minimization applies a DPO-style objective to align with preferences. The selection of visual-agnostic tokens aims to discourage spurious text-only shortcuts and force grounding in visual evidence. The method further adds a frequency-domain regularizer (“spectral preference alignment”, SPA). The authors evaluate primarily on AMBER, MMHal**, **POPE, and OBJHal, reporting consistent hallucination reductions at 7B/13B scales and favorable ablations for the three components: TP (token perturbation), CAS (cross-modal alignment score) for token selection, and SPA.

**Strengths:**

1. **Clear illustration on problem formulation.** Casting preference learning as a min–max problem over token-level perturbations helps mitigate the “style/phrase” overfitting often seen with vanilla DPO in multimodal settings.
2. **Broad evaluation on recognized hallucination suites.** Benchmarks (AMBER, MMHal, POPE, OBJHal, CHAIR) are aligned with current community practice for LVLM hallucination assessment.

**Weaknesses:**

1. **Why token perturbation?** Can the authors provide the reason / intuition that supports this design? i.e., any intuition to illustrate the rationality of this design? Moreover,
   1. What is the operational space of the perturbation? the whole vocabulary?
   2. Can the authors give some examples for what the text input ***after*** perturbation be like?
2. **Spectral regularization using FFT.** What is the point of using FFT to do alignment? What is the intuition / reason that supports this design? It's not enough to just say "Enforcing finegrained constraints may reintroduce spurious correlations that our min–max strategy aims to mitigate" (line 240-242).
3. **Frequency-domain regularizer needs sharper theoretical footing.**
    SPA is motivated as smoothing over local token perturbations, but the connection between **Fourier-space alignment** and *down-weighting spurious token-level correlations* remains largely unclear. A small theoretical or diagnostic study (e.g., spectrum-response correlation to CHAIR/POPE error modes) would clarify *why* SPA helps beyond acting as another consistency loss.
4. **Evaluation benchmarks & baselines.** Apart from existing benchmarks adopted in the manuscript, there are also some more comprehensive benchmarks for hallucinations such as HallusionBench [1] and MMStar [2]. Also, to validate the effectiveness of TARS, some newest multimodal DPO baseline methods should be considered in experiments, such as SymMPO [3] and HSA-DPO [4].

If the authors can solve my concerns, I am willing to increase the rating.

References:

[1] Guan et al., "HallusionBench: An Advanced Diagnostic Suite for Entangled Language Hallucination and Visual Illusion in Large Vision-Language Models" In CVPR 2024.

[2] Chen et al., "Are we on the right way for evaluating large vision-language models?" In NeurIPS 2024.

[3] Liu et al., “Mitigating Hallucination Through Theory-Consistent Symmetric Multimodal Preference Optimization” In NeurIPS 2025.

[4] Xiao et al., "Detecting and Mitigating Hallucination in Large Vision Language Models via Fine-Grained AI Feedback" In AAAI 2025.

**Questions:**

1. **Clarity/details.**
   - The paper defines **Sim((\phi(q), q))** (Eq. 7) but does not fully specify the similarity space or substitution strategy / search heuristic that  constructs (\phi)?
   - SPA (spectral regularizer) is introduced at a high level; please expand its exact form, hyper-parameters, and interaction with the DPO loss.
2. **SPA details:** What is the **exact spectral loss** and how sensitive is performance to its coefficient?

---

### Note · Authors · 2025-11-12

I have read and agree with the venue's withdrawal policy on behalf of myself and my co-authors.